# Controlling the Performance of Polymer Lasers via the Cavity Coupling

**DOI:** 10.3390/polym11050764

**Published:** 2019-05-01

**Authors:** Shuai Zhang, Junhua Tong, Chao Chen, Fengzhao Cao, Chengbin Liang, Yanrong Song, Tianrui Zhai, Xinping Zhang

**Affiliations:** Institute of Information Photonics Technology, College of Applied Sciences, Beijing University of Technology, Beijing 100124, China; zhangshuai@emails.bjut.edu.cn (S.Z.); jhtong@emails.bjut.edu.cn (J.T.); s201706083@emails.bjut.edu.cn (C.C.); wincfz@163.com (F.C.); s201706086@emails.bjut.edu.cn (C.L.); yrsong@bjut.edu.cn (Y.S.); zhangxinping@bjut.edu.cn (X.Z.)

**Keywords:** DFB polymer lasers, cavity coupling, azimuthally polarized output, lasing threshold

## Abstract

The polarization and threshold of distributed feedback (DFB) polymer lasers were controlled by adjusting the cavity coupling. The cavity of DFB polymer lasers consisted of two gratings, which was fabricated by a two-beam multi-exposure holographic technique. The coupling strength of the cavity modes was tuned by changing the angle between the two gratings. The threshold of the polymer lasers decreased with reducing the coupling strength of the cavity modes. A minimum threshold was observed at the lowest coupling strength. Moreover, the azimuthally polarized output of the polymer lasers was modified by changing the cavity coupling. These results may provide additional perspectives to improve the performance of DFB polymer lasers.

## 1. Introduction

In the last several decades, polymer lasers have attracted much interest and distributed feedback (DFB) polymer lasers have achieved many unique developments and applications [1,2,3,4]. DFB polymer lasers are the most promising polymer lasers due to the directionality and low loss of emission, long gain lengths, and high optical efficiency [5,6,7,8]. Due to the good confinement, two-dimensional (2D) DFB polymer lasers have a lower threshold and higher photoluminescence (PL) efficiency and the output beams have different profiles than one-dimensional (1D) DFB polymer lasers [9]. A variety of fabrication schemes have been demonstrated to construct the DFB cavities, such as interference lithography [10], nanoimprint lithography [11], electron beam lithography [12], and reactive ion etching [13]. Recently, considerable research efforts have been devoted to the special emission spot of 2D DFB polymer lasers, which is axisymmetric polarization light [14,15,16,17,18,19,20]. Axisymmetric polarizations include azimuthal and radial polarizations [21]. Azimuthally polarized light can be formed by intermodulation of two linear polarized lights. Additionally, the low threshold of the 2D DFB polymer laser is attributed to a better confinement of light than that of the 1D DFB polymer laser. In fact, the threshold behavior of 2D DFB polymer lasers can be explained quantitatively by a cavity coupling theory. The coupling strength can be tuned continuously by changing the angle between the substructures.

In this paper, the azimuthally polarized mode of 2D DFB polymer lasers was investigated experimentally and theoretically. 2D gratings were fabricated by a two-beam multi-exposure holographic technique. The fluorene-based polymer was spin-coated onto the grating structure, forming a laser device. The output characteristics of 2D DFB lasers were studied. A comparison of the experimental performances has been made based on different coupling effects in the DFB polymer lasers. The relationship between the lasing threshold and coupling coefficient in the DFB laser has been extracted and proved by the experiment. The threshold of the 2D polymer laser decreased when the cavity coupling strength was weakened. The azimuthally polarized output of the polymer lasers was adjusted by changing the cavity coupling.

## 2. Structures and Methods

The photoresist (PR, AR-P3170, Strausberg, Germany) grating was fabricated on a glass substrate (18 × 18 × 1 mm) by holographic technique. The effective area of the PR grating is about 100 mm^2^. The PR was first spin-coated onto a glass substrate to form a thin film at a speed of 2500 rpm. The samples were heated at 110 °C for 1 min by a dryer. After this, the prepared samples were exposed for holographic technique using a 343 nm pulsed laser (FLARE NX, Coherent, Santa Clara, CA, USA). The period (Λ) of the DFB cavity is determined by Λ = λ/(2sin*θ*), where λ is the wavelength used in the lithography process and *θ* is the angle between the two beams in interference lithography. The period of the gratings was 335 nm (*θ* = 31 degree). The peak height of the grating was around 170 nm. Different 2D gratings were fabricated by changing the angle between the two exposures. A rotatable sample-holder was designed, which can accurately regulate the different angles between two exposures in the two-beam multi-exposure holographic technique to form the 2D DFB cavities. In this experiment, the 2D DFB cavities with different angles were obtained. Due to the similar time of the two exposures used in the lithographic process, the periods in different directions were roughly equal. After exposure for 15 s, the grating structures were fabricated using a developer (AR-300-47, Allresist, Strausberg, Germany) for 5 s to dissolve the imprinted parts. The exposure and dissolving time were adjusted along with the various structural parameters. Later, poly [(9,9-dioctylfluorenyl-2,7-diyl)-alt-co-(1,4-benzo-(2,10,3)-thiadiazole)] (F8BT, Sigma-Aldrich, St. Louis, MO, USA) was dissolved in xylene at a concentration of 23.5 mg/mL. The F8BT solution was spin-coated onto the DFB cavities at a speed of 1800 rpm, forming a DFB polymer laser.

Figure 1a–e indicates the scanning electron microscopy (SEM) pictures of different kinds of gratings that were fabricated by holographic technique. Figure 1f–j shows the corresponding simulation results for different grating structures. The MATLAB simulation was based on the theory of two-beam multi-exposure holographic technique. The maximum and minimum values of the intensity distribution of the interference are indicated by red and blue, respectively. A spectrum analyzer from an optical spectrometer (USB 4000, Ocean Optics, FL, USA) was used to measure the optical extinction spectrum. The extinction spectrum of the different structures is shown in Figure 1k. For the transverse magnetic (TM) and transverse electric (TE) waves, there was a very small difference between the extinction peaks. The difference can be attributed to the variance of the effective refractive index of the cavity. The exaction peak resulted from the diffraction of the gratings, which influenced the lasing wavelength significantly. This will be discussed in detail later.

The polarization vectors of the output beam are *E*_1_ and *E*_2_. The effects of modulation from two gratings are defined as two matrixes $G_{1}\left(x,y\right)$ and $G_{2}\left(x,y\right)$. The matrixes are related to the geometric parameters of the two gratings [22,23].
(1)$$\{\begin{array}{l} E_{1}^{\prime }=G_{2}\left(x,y\right)E_{1}\\ E_{2}^{\prime }=G_{1}\left(x,y\right)E_{2} \end{array}$$

The polarization vectors of the output beam are modified by $G_{1}\left(x,y\right)$ and $G_{2}\left(x,y\right)$. The polarization vectors can be described by the Jones vector as:
(2)$$\left[\begin{array}{c} E_{x}^{\prime }\left(x,y\right)\\ E_{y}^{\prime }\left(x,y\right) \end{array}\right]=jexp\left(j\delta \right)\left(\begin{array}{c} -sin\delta \\ cos\delta \end{array}\right)$$
where δ is the angle between the two grating vectors. $E_{x}^{\prime }\left(x,y\right)$ and $E_{y}^{\prime }\left(x,y\right)$ are the vectors’ *x* and *y* components in the same coordinate system. The polarization orientation is determined by the grating direction. Figure 2 shows the schematic of the azimuthally polarized output of 2D DFB polymer lasers. The gratings fabricated by multi-exposures were mutually perpendicular. The polarization would be turned when mutual modulation effect was aroused by the other grating in the same device. The polarization vectors were turning to form a group of circular vectors. Therefore, the lasing spots of the 2D DFB polymer lasers were a hollow circle under the influence of the circular vectors.

## 3. Results

As shown in Figure 3a–c, the simulations of the lasing spots show an annular profile. In the experiment, a 400 nm femtosecond laser with a pulse duration of 200 fs and a repetition frequency of 1 kHz was employed as a pump source. The pump beam with a radius of 1.9 mm impinged on the sample surface with an incident angle of 50 degree. A neutral density filter was used to adjust the pump power. The laser spots of 2D polymer lasers were investigated systematically based on different DFB cavities. Figure 3d–f shows the lasing spots of different 2D DFB cavities. Figure 3g demonstrates a three dimensional energy distribution of the lasing spot. The spots were measured by the laser beam profiling system (Ophir-Spiricon LAB-USB-SP620, Logan, UT, USA). To verify the azimuthal polarization property of the output beam, a linear polarizer was set between the device and beam analyzer. A double lobed beam was observed when the output beam passed through a linear polarizer, as shown in Figure 3h,i. Obviously, the output beam was azimuthally polarized.

The azimuthally polarized mode is supported by the coherent combination of resonant fields [24,25]. In the 2D DFB cavity, two resonant fields are supported by the two gratings, respectively. The coherence of the two resonant fields varies with the angle between the two gratings. The coherence of two resonant fields are defined by the coupling coefficient of the 2D DFB cavity, as shown in Equation (3). The azimuthally polarized mode can be modified by changing the angle between two gratings. Thus, the profile of the lasing spots changes with the angle of the two gratings.

The absorption, photoluminescence and amplified spontaneous emission (ASE) spectra of F8BT were shown in Figure 4a. Figure 4b indicates the relationship between the threshold and the coupling coefficient as a function of the angle between two gratings. The experimental data (triangular points) are the numerical average value of the threshold of five samples under the same experimental condition. The labels ①–⑤ demonstrate different DFB cavities. The angle between two gratings decrease will enhance the coupling strength. The coupling strength greatly influences the threshold of DFB polymer lasers. Based on coupled-mode theory, the simulation results show good coherence with the experiment results. The coupling strength is regarded as a kind of “loss” that increases the laser threshold [26,27,28]. The coupling coefficient is used to express the coupling strength of cavities in the same plane of the devices. The 2D cavity is regarded as the combination of two 1D cavities (A and B). The mode energy of cavity A coupled to cavity B means the mode energy of cavity A decreases. The mode energy loss will result in an increase of the laser threshold, and vice versa. The mode coupling strength is determined by the coupling coefficient κ.

(3)$$\{\begin{array}{c} \frac{da_{1}}{dt}=-i\omega_{1}a_{1}+i\kappa a_{2}\\ \frac{da_{2}}{dt}=-i\omega_{2}a_{2}+i\kappa a_{1} \end{array}$$
where, $a_{1}$ and $a_{2}$ are the total intensity of coupling cavity, and κ is the coupling coefficient. $\omega_{0}$ is the lasing wavelength. $\omega_{1}$ and $\omega_{2}$ are the angular frequencies in different resonators. In our experiment, $\omega_{1}=\omega_{2}$. Equation (3) indicates the intermodulation of laser modes in a 2D DFB cavity. After arranging,
(4)$$\left[\begin{array}{cc} sin\theta \cdot \omega_{0}-\omega & -\kappa \\ -\kappa & \left(1+cos\theta \right)\cdot \omega_{0}-\omega \end{array}\right]=0$$

Equation (4) is deduced. θ is the angle between the two gratings in the 2D cavities, ranging 0–90 degree. ω will be solved to yield κ(θ). The function between the angle of two gratings θ and the coupling coeicient κ is obtained. Thus, the coupling coefficient κ can be adjusted by changing the angle θ. Obviously, orthogonal cavities produced the smallest coupling effect. In other words, it means the weakest coupling strength. From Figure 4b, the threshold of 2D DFB polymer lasers increases with the decrease of angle θ. The emission spectra of different polymer lasers are plotted in Figure 4b, which are measured using an optical spectrometer (Maya 2000 Pro, Ocean Optics, FL, USA). The insets in Figure 4b represent the corresponding DFB cavities. In the experiment, the thickness of the polymer waveguide is about 130 nm, so only the fundamental propagating mode (TE_0_) can be excited in the polymer film. Then the TE_0_ mode is Bragg-scattered out by the grating structure, as shown in Figure 1k. The slight difference between the lasing wavelength and the diffraction peak in Figure 1k is attributed to the low accuracy of interference lithography. As shown in Figure 4d, the lasing threshold of the cavity with different angle θ of 0, 30, 45, 60, and 90 degrees are 62.7, 55.8, 49.8, 46.6, and 40.3 μJ/cm^2^, respectively. Note that the periods of the two grating are same in the 2D cavity, so the loss of the mode mismatch is small for the whole laser device. The loss of the laser system is mainly caused by the material absorption. The threshold of the laser device will increase significantly when the periods of the two gratings are different, due to the loss caused by the mode mismatch.

Figure 5 shows the threshold of 2D DFB cavities with two different gratings. The periods of two gratings are 335 and 340 nm, respectively. The insets in Figure 5b show the lasing spectra of different 2D DFB polymer lasers. The calculated coupling coefficient agrees well with the measured thresholds. Note that when the cavity period is too large or too small, the resonance wavelength of the cavity shifts away from the peak of the gain spectrum. The laser threshold will increase significantly and no lasing may be able to occur. In our experiment, the 340 nm grating is too large to support lasing. Thus, the 340 nm grating acts as a loss channel of 335 nm grating due to the mode mismatch. The physical mechanism of DFB lasers has been explained in the published paper [29]. As a result, the threshold of the laser device is noticeably increased, as shown in Figure 5. Especially, the lasing threshold increased from 40.3 to 150 μJ/cm^2^ for the orthogonal case when the period of one grating changed from 335 to 340 nm. This can be attributed to the fact that only one grating structure supports lasing for the DFB cavity with two different gratings. Moreover, the lasing threshold of the 2D DFB polymer lasers in Figure 5 is higher than that of the 1D DFB polymer laser in Figure 4 due to the additional loss channel. Note that the relationship between the laser threshold and the angle of two grating is unchanged.

## 4. Conclusions

In summary, the performance of 2D DFB polymer lasers was controlled by changing the coupling strength between the two substructures in the cavity. The coupling strength of the cavity was tuned by changing the angle between the two substructures, which was achieved by using a two-beam multi-exposure holographic technique. The profile of the azimuthally polarized output was tuned by changing the cavity coupling. Furthermore, the relationship between the laser threshold and the cavity coupling was deduced, which agreed well with the experimental results. The cavity coupling acted as an additional loss channel if the periods of the two substructures were different, which resulted in a higher threshold. The results may be utilized in the design of high-performance polymer lasers.

## Figures and Tables

**Figure 1 polymers-11-00764-f001:**
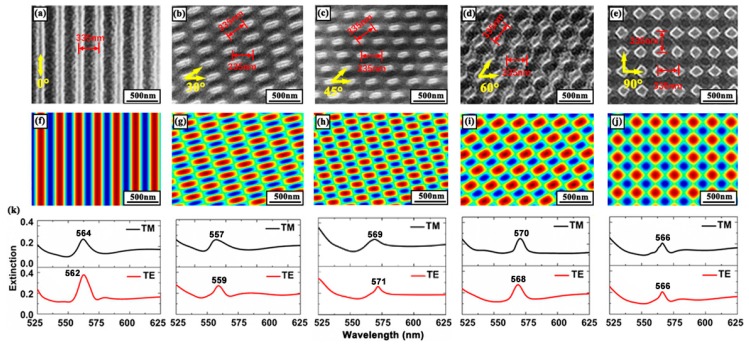
SEM images of (**a**) a 1D distributed feedback (DFB) grating and (**b**–**e**) 2D DFB gratings. The period is 335 nm. (**f**–**j**) Simulations of the interference pattern. (**k**) The transverse magnetic and transverse electric waveguide modes of different DFB gratings.

**Figure 2 polymers-11-00764-f002:**
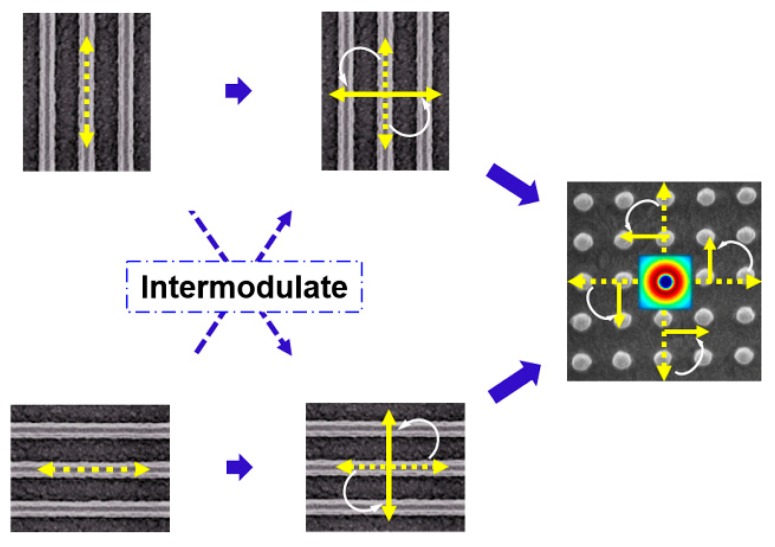
Schematic of the polarization intermodulation of laser modes in a 2D DFB cavity. The dotted double-arrow indicates the polarization of the DFB polymer laser. The solid double-arrow indicates the polarization of the DFB polymer laser after intermodulation. The white arrow denotes the rotation of the polarization.

**Figure 3 polymers-11-00764-f003:**
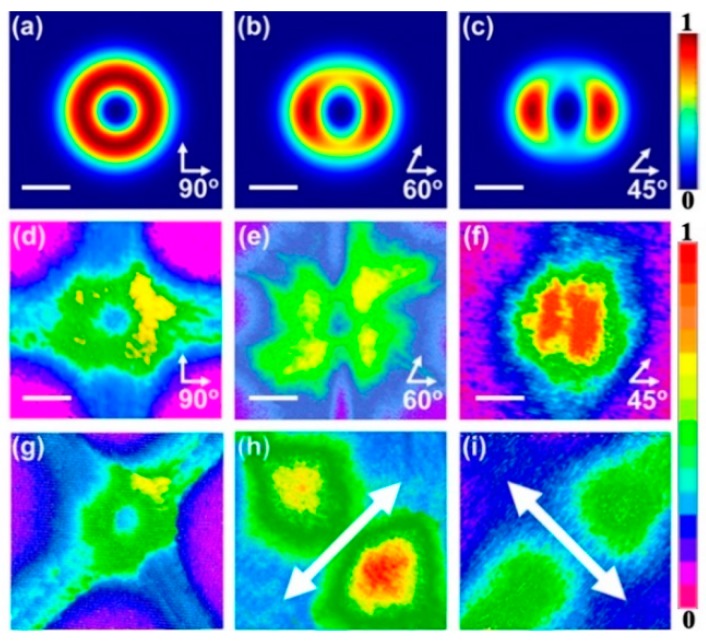
Profiles of the output beam of the DFB polymer laser. (**a**–**c**) Simulation results of different laser spots. (**d**–**f**) Images of different laser spots. Angles formed by white arrows represent the direction of the substructure in the 2D cavity. The scale bar is 0.3 μm. (**g**) Three-dimensional energy distribution of the lasing spot. (**h**,**i**) Laser spots measured through a linear polarizer. The double-head arrow indicates the direction of the polarizer.

**Figure 4 polymers-11-00764-f004:**
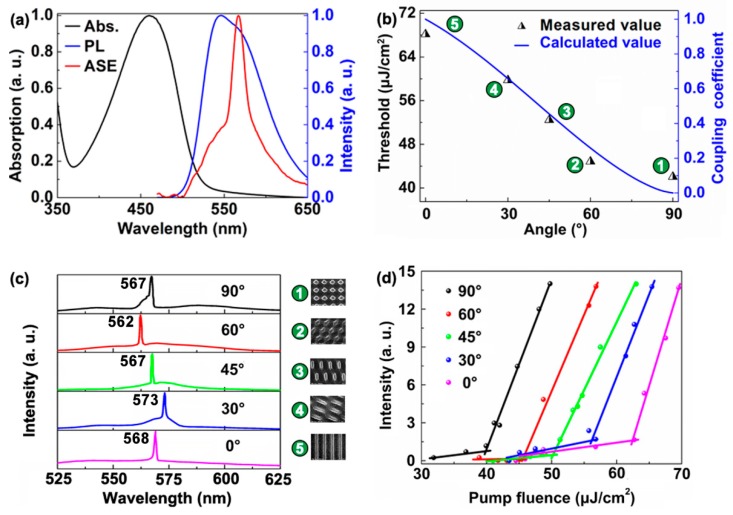
(**a**) Absorption, photoluminescence (PL), and amplified spontaneous emission spectra of F8BT. (**b**) Relationship between the coupling coefficient and the threshold of the 2D DFB cavity. The triangles indicate the average threshold for different 2D DFB polymer lasers. (**c**) Measured spectra of 2D DFB polymer lasers. (**d**) Output intensity of 2D DFB polymer lasers as a function of the pump fluence.

**Figure 5 polymers-11-00764-f005:**
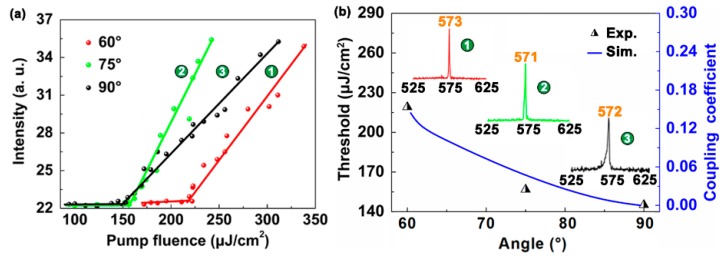
Thresholds of the 2D DFB polymer lasers with different periods. (**a**) Output intensity of 2D DFB polymer lasers as a function of the pump fluence. (**b**) Relationship between the angle *θ* and the threshold of 2D DFB cavities. The triangles indicate the measured threshold for different 2D DFB polymer lasers. The blue curve denotes the calculated value of the coupling coefficient.

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
