# Peer review of "Controlling the Performance of Polymer Lasers via the Cavity Coupling"

_polymers, 2019, doi:10.3390/polym11050764_

Round 1

Reviewer 1 Report

This is a well-written manuscript, and I am happy to recommend publication in Polymers.  My only 'critique' is that the present tense should be used when describing the results.

Author Response

COMMENTS:

1) Moderate English changes required

Response:

Many thanks! We have checked the manuscript carefully.

Comments and Suggestions for Authors

2) This is a well-written manuscript, and I am happy to recommend publication in Polymers. My only 'critique' is that the present tense should be used when describing the results.

Response:

We really appreciate that you read our manuscript carefully and affirm our work. As your suggestion, we have revised the manuscript carefully.

Special thanks to you for your valuable comments! 

Reviewer 2 Report

The authors have fabricated a 1D DFB resonator by using holographic lithography and 2D DFB resonators by using two-beam multi-exposure holographic technique. The holographic gratings have been exposed over photoresist and a chemical development process was used (SEM images of some of the gratings are reported).  A conjugated polymer (F8BT) was used as a gain medium. Polarization and efficiency of DFB emission devices are discussed theoretically and experimentally by changing the grating geometry. A complete study is performed to study the cavity coupling of 2D DFBs. However, there are certain ambiguities that can make it difficult to communicate the results. Finally, remarking in the text the main original contribution with respect the state of the art related with this work, could improve the quality and value of this manuscript.

1) A more detailed description of all DFB cavities shown and discussed in Figure 1, 3, 4 and 5 could improve the work understanding. At experimental section we can only find a description of a 2D grating with 335 nm period in both directions, but it is not clear by seeing SEM images that the periods of the two gratings were identical (line 56, 163). Please, could you provide the values of periods in both directions and their errors from SEM images or MATLAB simulations (line 162)?

2) It is assumed identical angular frequencies in the different resonators of the experiment (line 144, equation 4). Could be this statement confirmed or re-evaluated with the corresponding error (line 158)? Could be commented the resolution of the spectrometer used to obtain the spectra in Figure 4? Which is the linewidth of the different spectra measured? The physical mechanism which explain 1D and 2D DFB emission it is not explained in this work, could be explained and/or cited some references? For example, you can review: Optical Materials Express, Vol. 7(4), 1295.

3) At the end of section 3, it is shown some experimental results of 2D DFB with different grating periods (335 and 340 nm). Why is assumed that no surface-emitting modes can be excited by the 340 nm grating? Why is not provided similar simulations as presented with 2D DFB gratings of similar periods? Could be provided a detailed explanation? What is the thickness of the active film of these DFBs (with different periods)? What is the cut-off thickness of the device? Absorption, Photoluminescence, Amplified Spontaneous Emission spectrum, effective refractive index, gain or losses measurements could be provided for a better understanding?

-Line 9: Is it the polarization and threshold really controlled by adjusting the cavity coupling?

-Line 10: 1D DFB grating shown in Figure 1 and 5 is fabricated by a two-beam multi-exposure holographic technique?

-Line 20: Could you comment the unique developments referenced in ref 1 to 6?

-Line 22 to 23: On this work, the threshold of 2D DFB shown in Figure 5 is higher than 1D DFB shown in Figure 4. Please could you detail the differences between the fabrication of DFB number 5 of Figure 4 and the rest of DFBs? Is it the DFB resonator number 5 a 1D DFB, a substructured DFB or a 2D DFB? Is it fabricated with one or two exposures?

-Line 25: The reference 12 is related with a grating fabricated by direct writing (interference ablation). Please, could you develop with more detail the holographic lithography technique and include some references?

-Line 46: What is the thickness of each thin film presented in the manuscript (line 155)?

-Line 49: It is a pulsed or a continuous laser?

-Line 51: Could be interference lithography misunderstood with the technique used in ref 12?

-Line 51: Period of gratings used in Figure 5 is not detailed in this section.

-Line 51 and 60: Are both descriptions similar between them?

-Line 52: In which plane are the angle of the exposure changed? Is it applied the same plane in DFB (a) as the rest of DFBs of figure 1?

-Line 56, 58: Could you provide the time? The period depended on the angle between interference beams (line 49).

Line 158, 162: Are they consistent between them?

-Line 77: Is it the effective refractive index obtained?

-Line 102: Pulse duration, frequency…?

-Line 103: Could you provide the diameter of the spot or the diameter of the incident ellipse? Please, check Heliotis 2004 for a better understanding of the shape spot influence.

-Figure 3: Image (g) is not mentioned on the main manuscript. Which is the main contribution of this figure with respect the published by Heliotis et. al. in 2004?

-Figure 4b: Is the linewidth of DFB 1 and 4 according to a DFB spectrum? Was the data of linewidth vs pump fluence measured?

-Figure 4c: Slope efficiency values is not obtained but it is discussed in the text (line 161?).

-Line 189: The state of the art of high-performance DFB lasers has not been previously cited in the manuscript. The state of the art of F8BT polymer laser was not discussed previously in the text.

Author Response

COMMENTS:

1) English language and style are fine/minor spell check required

Response:

Many thanks! We have checked the manuscript carefully.

Comments and Suggestions for Authors

2) The authors have fabricated a 1D DFB resonator by using holographic lithography and 2D DFB resonators by using two-beam multi-exposure holographic technique. The holographic gratings have been exposed over photoresist and a chemical development process was used (SEM images of some of the gratings are reported). A conjugated polymer (F8BT) was used as a gain medium. Polarization and efficiency of DFB emission devices are discussed theoretically and experimentally by changing the grating geometry. A complete study is performed to study the cavity coupling of 2D DFBs. However, there are certain ambiguities that can make it difficult to communicate the results. Finally, remarking in the text the main original contribution with respect the state of the art related with this work, could improve the quality and value of this manuscript.

Response:

We really appreciate that you read our manuscript carefully and made these valuable comments which are of great help to improve this paper. We have presented the main original contribution in the revised version.

Our main contribution includes two points. We revealed the relationship between the threshold of different polymer lasers and the coupling strength of the cavity modes. We found the azimuthally polarized output of the polymer lasers was adjusted by changing the cavity coupling.

We have emphasized this point in the revised manuscript. (Lines 42-44)

3) A more detailed description of all DFB cavities shown and discussed in Figure 1, 3, 4 and 5 could improve the work understanding. At experimental section we can only find a description of a 2D grating with 335 nm period in both directions, but it is not clear by seeing SEM images that the periods of the two gratings were identical (line 56, 163). Please, could you provide the values of periods in both directions and their errors from SEM images or MATLAB simulations (line 162)

Response:

Many thanks! We are very sorry for the misleading of the inaccurate presentation. We have modified Fig. 1. We have added some detailed description in the revised version.

As shown in Fig. R1, the grating period have been denoted by double-head arrows. All periods are 335 nm.

The grating period (Λ) is determined by Λ=λ/(2sinθ), where λ is the wavelength and θ is the angle between the two beams in interference lithography. So, the grating period is determined by the angle θ for a fixed wavelength (335 nm). In the experiment, θ was fixed to 31 degree. So, all grating periods were exactly same in the SEM images or MATLAB simulations.

Figure R1. SEM images of (a) a 1D DFB grating and (b)-(e) 2D DFB gratings. (f)-(j) Simulations of the interference pattern. (k) The transverse magnetic and transverse electric waveguide modes of different DFB gratings

The figure has been replotted in the manuscript (Fig. 1 on page 3).

4) It is assumed identical angular frequencies in the different resonators of the experiment (line 144, equation 4). Could be this statement confirmed or re-evaluated with the corresponding error (line 158)? Could be commented the resolution of the spectrometer used to obtain the spectra in Figure 4? Which is the linewidth of the different spectra measured? The physical mechanism which explain 1D and 2D DFB emission it is not explained in this work, could be explained and/or cited some references? For example, you can review: Optical Materials Express, Vol. 7(4), 1295.

Response:

The frequencies in the different resonators are decided by the wavelength of laser modes. For the 2D cavity with two same gratings, the laser modes are same due to the same grating period. So, only one laser wavelength (angular frequency) can be observed in the 2D cavity with two same gratings in the experiment, as shown in Fig. 4. So, the angular frequencies in the different resonators are identical in the experiment and simulations.

The resolution of optical spectrometer (Maya 2000 Pro, Ocean Optics) is 0.5 nm. In Fig. 4(b), the linewidths are about 1.2 nm, 1.1 nm, 0.8 nm, 1.3 nm, 1.1 nm, respectively.

Thank you for your valuable suggestion. As you suggested, we added a new reference (Optical Materials Express, Vol. 7(4), 1295).

We have explained this point in the revised version. (Line 191, 288-290)

5) At the end of section 3, it is shown some experimental results of 2D DFB with different grating periods (335 and 340 nm). Why is assumed that no surface-emitting modes can be excited by the 340 nm grating? Why is not provided similar simulations as presented with 2D DFB gratings of similar periods? Could be provided a detailed explanation? What is the thickness of the active film of these DFBs (with different periods)? What is the cut-off thickness of the device? Absorption, Photoluminescence, Amplified Spontaneous Emission spectrum, effective refractive index, gain or losses measurements could be provided for a better understanding?

Response:

For DFB polymer lasers, no lasing occurs if the cavity period isn't proper. When the cavity period is too large or too small, the resonance wavelength of the cavity shifts away from the peak of the gain spectrum. So, the laser threshold will increase significantly. Even no lasing can occur. In our experiment, the 340 nm grating is too large to support lasing.

Many thanks! We have added a similar simulation as presented with 2D DFB gratings of similar periods. The relationship between the laser threshold and the angle of two grating is unchanged.

Figure R2. Thresholds of the 2D DFB polymer lasers with different periods. (a) Output intensity of 2D DFB polymer lasers as a function of the pump fluence. (b) Relationship between the angle θ and the threshold of 2D DFB cavities. The triangles indicate the measured threshold for different 2D DFB polymer lasers. The blue line is the calculated value of coupling coefficient.

Figure R3 showed SEM images of the cross section of the sample (with different periods). It can be seen that the thickness of active film and the PR grating are 130 nm and 170 nm, respectively. The total thickness of the waveguide is 300 nm, which determines the cut-off frequency of the laser device. It will be discussed in detail later.

Figure R3. SEM images of the cross section of the sample.

For the cut-off thickness of the device

According to the relationship of the waveguide (WG) mode with the thickness, if the thickness (d) of the polymer film is greater than or equal to ds or dp (which are the cut-off thicknesses for the existence of WG modes), additional waveguide modes can be excited. For the mth (ms=0,1,2,...) order s-polarized mode (TEm), ds is given by

                                                       (1)

for the mth (mp=1,2,3,...) order p-polarized mode (TMm), dp is given by

                                                         (2)

where εe is the effective refractive dielectric constant and λ is the emission wavelength. In our work, for εe= 1.69 (spectroscopic ellipsometer, ESNano, Ellitop), λ=571 nm, the cut-off thickness for TE0 and TE1 modes are 171 nm (<300 nm) and 512 nm (>300 nm), respectively. Similarly, the critical thickness for TM1 mode is 341 nm (>300 nm). In our experiment, the thickness of active film and the PR grating are 130 nm and 170 nm, respectively. The total thickness of the waveguide is 300 nm. Thus, only TE0 mode can be excited.

Many thanks! We have added the absorption, photoluminescence, amplified spontaneous emission spectrum in Fig. R4(a).

Figure R4 (a) Absorption, photoluminescence and amplified spontaneous emission spectra of F8BT. (b) Relationship between the coupling coefficient and the threshold of the 2D DFB cavity. The triangles indicated the average threshold for different 2D DFB polymer lasers. (c) Measured spectra of 2D DFB polymer lasers. (d) Output intensity of 2D DFB polymer lasers as a function of the pump fluence.

We have added several sentences to explain this point in the revised version. (Lines 53-54, 136-137, 174-175, 189-192, Fig. 4 on page 5)

6) -Line 9: Is it the polarization and threshold really controlled by adjusting the cavity coupling?

Response:

The answer is positive. The polarization and threshold can be controlled by adjusting the cavity coupling.

7) -Line 10: 1D DFB grating shown in Figure 1 and 5 is fabricated by a two-beam multi-exposure holographic technique?

Response:

One dimensional DFB gratings are fabricated by two-beam holographic technique with one exposure. Other 2D DFB gratings are fabricated with two exposures.

8) -Line 20: Could you comment the unique developments referenced in ref 1 to 6?

Response:

Many thanks! In Ref. 1, conjugated polymer lasing was firstly reported in the microcavities. In Ref. 2, the authors reviewed the development of organic semiconductor lasers. In Ref. 3, the authors reviewed the development of applications of polymer lasers. In Ref. 4, the physical mechanism of distributed-feedback polymer lasers were investigated.

We have removed two references (Refs. 5 and 6) because they are redundant.

We have added several sentences to explain this point in the revised version. (Lines 19-21, 218-228)

9) -Line 22 to 23: On this work, the threshold of 2D DFB shown in Figure 5 is higher than 1D DFB shown in Figure 4. Please could you detail the differences between the fabrication of DFB number 5 of Figure 4 and the rest of DFBs? Is it the DFB resonator number 5 a 1D DFB, a substructured DFB or a 2D DFB? Is it fabricated with one or two exposures?

Response:

In Fig. 4, the periods of the two grating are same in the 2D DFB lasers, so the loss of the mode mismatch was negligible for the whole laser device. So, the loss of the laser system is mainly caused by the material absorption. However, there is the mode mismatch in the 2D DFB polymer lasers with different periods. The loss caused by the mode mismatch leads a high threshold.

The lasing threshold increased from 40.3 to 150 μJ/cm2 for the orthogonal case when the period of one grating changed from 335 nm to 340 nm. It can be attributed that only one grating structure supports lasing for the DFB cavity with two different gratings. Moreover, the lasing threshold of the 2D DFB polymer lasers in Fig. 5 is higher than that of the 1D DFB polymer laser in Fig. 4 due to the additional loss channel.

Yes, the difference between them is that Fig. 4(a) 1D DFB gratings use one exposure and 2D gratings use two exposures.

We have added one sentence to explain this point in the revised version. (Lines 195-197)

10) -Line 25: The reference 12 is related with a grating fabricated by direct writing (interference ablation). Please, could you develop with more detail the holographic lithography technique and include some references?

Response:

Very sorry for the mistake! Interference ablation technique is different from holographic lithography technique. We have deleted the reference.

11) -Line 46: What is the thickness of each thin film presented in the manuscript (line 155)?

Response:

We presented the thickness of polymer film is 130nm in line 167 and the thickness PR film is 170nm in line 53.

We have added one sentence to explain this point in the revised version. (Lines 54, 164)

12) -Line 49: It is a pulsed or a continuous laser?

Response:

It is a pulsed laser.

We have explained this point in the revised version. (Line 50)

13) -Line 51: Could be interference lithography misunderstood with the technique used in ref 12?

Response:

Sorry for the misleading! We have deleted the reference.

14) -Line 51: Period of gratings used in Figure 5 is not detailed in this section.

Response:

We have described the different period of gratings in line189.

15) -Line 51 and 60: Are both descriptions similar between them?

Response:

Many thanks! For clarify, we have deleted the sentence in the revised version (lines 62-63).

16) -Line 52: In which plane are the angle of the exposure changed? Is it applied the same plane in DFB (a) as the rest of DFBs of figure 1?

Response:

Figure R5 demonstrates the layout of interference lithography. The angle of the exposure changed in the plane perpendicular to the layout (paper surface) in Fig. R5. Yes, all of them are applied in the same plane.

Figure R5. Layout for interference lithography. Ri (i = 1,2,3,4) denotes the mirrors. BS denotes the beam-splitter.

17) -Line 56, 58: Could you provide the time? The period depended on the angle between interference beams (line 49).

Response:

As your suggestion, we have provided the exposure time, the dissolving time, and the angle between interference beams in line 59, 60, and 53, respectively.

18) -Line 158, 162: Are they consistent between them?

Response:

Yes. We have deleted the sentence for clarify in the revised version (lines 61-63).

19) -Line 77: Is it the effective refractive index obtained?

Response:

The effective refractive index is obtained by .  is the emission wavelength.  indicates the period. In our experiment, the effective refractive index is about 1.68 at 564 nm (=335 nm).

There was very small difference between the extinction peaks of TE and TM waves, which implies the effective refractive index difference is very small. It can be estimated by  (~0.003).

20) -Line 102: Pulse duration, frequency…?

Response:

Many thanks! We have added the information in the revised manuscript (line 107-108).

21) -Line 103: Could you provide the diameter of the spot or the diameter of the incident ellipse? Please, check Heliotis 2004 for a better understanding of the shape spot influence.

Response:

As you suggested, we have added the information in line 104-106.

22) -Figure 3: Image (g) is not mentioned on the main manuscript. Which is the main contribution of this figure with respect the published by Heliotis et. al. in 2004?

Response:

Many thanks! We have added one sentence to mention it.

Figures 3(e) and 3(f) are the main contribution of the figure. We found the azimuthally polarized output of the polymer lasers was adjusted by changing the cavity coupling. When the angle between two gratings changes from 90 to 0 degree, the annular spot will gradually be split into two parts. In Prof. Heliotis’s paper, the shape of the laser spot is annular and the angle between two gratings is fixed to 90 degree.

We have added the information in the revised manuscript (lines 109-110, 119-120).

23) -Figure 4b: Is the linewidth of DFB 1 and 4 according to a DFB spectrum? Was the data of linewidth vs pump fluence measured?

Response:

Yes. We have measured the data. Figure R6 showed the emission spectra of 2D DFB polymer lasers.

Figure R6. Emission spectra of polymer lasers based on 2D DFB cavities with two gratings. (a) The angle between two gratings is 30 degree. (b) The angle between two gratings is 90 degree.

24) -Figure 4c: Slope efficiency values is not obtained but it is discussed in the text (line 161?).

Response:

Our aim is to compare the relative value of the slope efficiency. In our experiment, all slope efficiency values are obtained under similar experimental conditions.

For clarify, we have deleted the sentence in the revised version (lines 169-172).

25) -Line 189: The state of the art of high-performance DFB lasers has not been previously cited in the manuscript. The state of the art of F8BT polymer laser was not discussed previously in the text.

Response:

As your suggestion, we have added several references in the introduction section to explain this point.

We have added the information in the revised manuscript (lines 19-21, 25, 242-244).

Special thanks to you for your helpful comments!

Round 2

Reviewer 2 Report

The manuscript has been improved, whereas I still find some  ambiguities in the new version. I explain here the ambiguities detected  in the cover letter:

4) The gratings shown in Figure 1 have the exactly same period as authors have confirmed. However, extinction (Fig 1k) and DFB emission peaks (Figure 4c) seems slightly different for each grating (but it is difficult to know exactly the peak wavelength showing the large interval of wavelength. May be a shorter interval 560-580 nm would be more clear. Therefore, taking into account that thickness and refractive index are the same, it seems that the Bragg law can not explain these results, is not it?

5) Absorption and Photoluminscence spectra match with previously published in Journal and Material Chemistry C “Ladder-type poly(indenofluorene-co-benzothiadiazole)s as efficient gain media for organic lasers: design, synthesis, optical gain properties, and stabilized lasing properties”. However ASE spectrum is different, please revise the measurements.

10) Please could you add any reference of holographic lithography to fabricate DFB resonators (line 6)?

19) If we asume that the Bragg law can be used to modelize the results. I have some doubts. Why the DFB peaks is closer to the Bragg wavelength calculated with a period of 340 nm? Please, revise the scale and the Interval of axis of the inset in Fig. 5c to clearly appreciate the DFB wavelength emission. Line 254, a peak at 571 nm would be inside of the gain (ASE) spectrum. Line 257, is the physical mechanism of the presented results explained in reference 28?

23) From  data of Figure R6 (cover letter) it is difficult to appreciate the spectra below theshold and the sudden decrease in the emission linewidth.

Author Response

Referee #2

COMMENTS:

1) English language and style are fine/minor spell check required

Response:

Many thanks! We have checked the manuscript carefully.

Comments and Suggestions for Authors

4) The gratings shown in Figure 1 have the exactly same period as authors have confirmed. However, extinction (Fig 1k) and DFB emission peaks (Figure 4c) seems slightly different for each grating (but it is difficult to know exactly the peak wavelength showing the large interval of wavelength. May be a shorter interval 560-580 nm would be more clear. Therefore, taking into account that thickness and refractive index are the same, it seems that the Bragg law cannot explain these results, is not it?

Response:

As your suggested, we have marked each peak wavelength in Fig. 1k and Fig. 4c. Now it is easy to know exactly the peak wavelength.

Yes, there is slightly different between the peak wavelength of Fig. 1k and Fig. 4c. Because the spectra are measured under different experimental conditions.

For the extinction measurement, a weak white light is employed as a source. For the emission measurement, a strong fs laser is employed as a pump source. All measurements are performed at room temperature and without cooling. For the latter, the thermal effect will slightly affect the effective refractive index of the laser device due to the absorption of a strong laser energy. Moreover, the inhomogeneity of the polymer/grating will affect the effective refractive index of the laser device, which is caused by the spin-coating method.

We have marked the peak wavelengths in the revised manuscript. (Fig. 1k and Fig. 4c)

5) Absorption and Photoluminscence spectra match with previously published in Journal and Material Chemistry C “Ladder-type poly(indenofluorene-co-benzothiadiazole)s as efficient gain media for organic lasers: design, synthesis, optical gain properties, and stabilized lasing properties”. However ASE spectrum is different, please revise the measurements.

Response:

Yes, there is a bit different between our results and the published paper (JMCC). It may be attributed to the difference of the molecular weight (around 15000-200000), the concentration (23.5 mg/ml VS 20 mg/ml), and the solvent.

The F8BT used in our experiment is purchase from Sigma-Aldrich. The measured spectra are consistent with the data provided by the company, as shown in Fig. R1.

Fig. R1. Absorption and PL spectra of F8BT.   (Download from the Sigma-Aldrich website)

10) Please could you add any reference of holographic lithography to fabricate DFB resonators (line 6)?

Response:

As you suggested, we have added a pioneer work (Phys. Rev. E 2003, 67, 056619) about fabricating microstructures by holographic lithography.

We have added the reference in the revised version. (Lines 26, 228-229)

19) If we asume that the Bragg law can be used to modelize the results. I have some doubts. Why the DFB peaks is closer to the Bragg wavelength calculated with a period of 340 nm? Please, revise the scale and the Interval of axis of the inset in Fig. 5c to clearly appreciate the DFB wavelength emission. Line 254, a peak at 571 nm would be inside of the gain (ASE) spectrum. Line 257, is the physical mechanism of the presented results explained in reference 28?

Response:

The effective refractive index of each DFB cavity is different due to the difference of the cavity parameters (include polymer/PR thickness, cavity type, substrate). It can be roughly estimated by the equation (.  is the emission wavelength.  indicates the period.) or measured by a spectroscopic ellipsometer. So, it is not intuitive to compare the peak wavelength of different samples with different effective index.

As you suggested, we have marked the peak wavelengths.

Yesthe answer is positive.

We have marked the wavelengths of the peaks in the revised manuscript. (Figure 5c)

23) From data of Figure R6 (cover letter) it is difficult to appreciate the spectra below theshold and the sudden decrease in the emission linewidth.

Response:

After adding the spectra below threshold (the black curve), it is easy to observe the sudden decrease in the emission linewidth, as shown in Fig. R2.

Fig. R2. Emission spectra of polymer lasers based on 2D DFB cavities with two gratings. (a) The angle between two gratings is 30 degree. (b) The angle between two gratings is 90 degree.
